# Enhanced Antitumor Activity of Korean Black Soybean Cultivar ‘Soman’ by Targeting STAT-Mediated Aerobic Glycolysis

**DOI:** 10.3390/antiox14020228

**Published:** 2025-02-18

**Authors:** Su Hwan Park, Jeong Hyun Seo, Min Young Kim, Hye Jin Yun, Beom Kyu Kang, Jun Hoi Kim, Su Vin Heo, Yeong Hoon Lee, Hye Rang Park, Man Soo Choi, Jong-Ho Lee

**Affiliations:** 1Department of Health Sciences, Dong-A University, Busan 49315, Republic of Korea; psh8316@gmail.com (S.H.P.); hyejin991011@gmail.com (H.J.Y.); 2Department of Southern Area Crop Science, National Institute of Crop Science, Rural Development Administration, Miryang 50424, Republic of Korea; kmyqwer@korea.kr (M.Y.K.); hellobk01@korea.kr (B.K.K.); itomi123@korea.kr (J.H.K.); hsb3937@korea.kr (S.V.H.); sky3832@korea.kr (Y.H.L.); hrpark6@korea.kr (H.R.P.); mschoi73@korea.kr (M.S.C.); 3Department of Biomedical Sciences, Dong-A University, Busan 49315, Republic of Korea

**Keywords:** black soybean, Soman, Seoritae, aerobic glycolysis, tumor growth

## Abstract

Black soybeans have numerous health benefits owing to their high polyphenolic content, antioxidant activity, and antitumor effects. We previously reported that the Korean black soybean cultivar ‘Soman’ possesses higher anthocyanin and isoflavone contents and superior antioxidant potential than other Korean black soybean cultivars and landraces (Seoritae) do. Here, we investigated and compared the antitumor effects of Soman and Seoritae and aimed to elucidate the possible mechanisms of action. Soman inhibited cancer cell proliferation and was more potent than Seoritae. Mechanistically, Soman inhibited the phosphorylation of the signal transducer and activator of transcription (STAT1, 3, and 5) in a reactive oxygen species (ROS)-independent manner, subsequently decreasing glycolytic enzyme expression and the activities of pyruvate kinase and lactate dehydrogenase. Thus, Soman suppressed glucose uptake, lactate production, and ATP production in cancer cells. Additionally, it inhibited tumor growth in a B16F10 murine melanoma syngeneic model, accompanied by reduced STAT1 phosphorylation and decreased proliferation in Soman-treated mice, more potently than observed in Seoritae-treated mice. These findings showed that Soman exerted superior antitumor activities by suppressing STAT-mediated aerobic glycolysis and proliferation. Overall, our findings demonstrate the potent, tumor-suppressive role of Soman in human cancer and uncover a novel molecular mechanism for its therapeutic effects in cancer treatment.

## 1. Introduction

Cancer is a leading cause of death worldwide and has a high mortality rate. In 2019, an estimated 23.6 million new cancer cases were reported, which increased from those in 2010 (18.7 million cases) and the 10.0 million cancer deaths reported globally, with an estimated 250 million disability-adjusted life years due to cancer [1]. Risk factors include smoking, unhealthy diets, and pollution, all of which increase the likelihood of developing cancer [2]. In recent years, various cancer treatment options have emerged, including surgery, radiation therapy, and chemotherapy. Approximately one-third of patients with cancer are estimated to be cured through surgery or radiation therapy, whereas chemotherapy is necessary for systemic treatment in cases of metastatic cancer [3]. However, these cancer treatment options can cause significant side effects and are relatively expensive. Therefore, it is essential to develop and discover new medicines from natural sources, particularly plant-derived functional foods, to ensure that these drugs are widely available and affordable [3].

The Warburg effect, a metabolic hallmark of cancer, is a phenomenon in which cancer cells primarily generate energy through aerobic glycolysis, regardless of extracellular oxygen availability [4]. As glycolysis is less efficient at producing ATP, cancer cells accelerate glycolytic flux by increasing glucose uptake, followed by lactate production in the cytosol to generate energy [5]. This metabolic alteration is essential for meeting the increased energy and biosynthetic demands of tumor cells and facilitating their rapid growth [6,7]. Therefore, targeting the Warburg effect is emerging as a promising cancer treatment strategy.

Soybeans (*Glycine max*) are a widely cultivated crop that has played a significant role in traditional diets and medicinal practices for thousands of years, especially in Asian cultures [8,9]. Their demand is increasing in diverse industries, such as food, pharmaceuticals, and biofuels [10]. Several studies have verified that soybeans are a rich source of essential amino acids, vitamins, and minerals, along with polyphenols (mainly anthocyanins, isoflavones, and phenolic acids), which are associated with various health benefits, including diabetic, cancer, and inflammatory activities and cardiovascular disease treatments [3,8,9,11,12]. Moreover, soybeans have distinct antioxidant properties that inhibit the overproduction of reactive oxygen and reactive nitrogen species in the human body through a free radical-quenching mechanism [3,13].

Different-colored soybeans have been reported worldwide, including yellow, black, brown, and green, which result from the presence of anthocyanins, chlorophyll, and other pigments in the seed coat [13,14]. Among them, black soybeans are gaining attention for their superior nutritional content, including high levels of protein and anthocyanins and excellent antioxidant effects compared to those of other colored soybeans [3,13,15,16,17]. Anthocyanins are water-soluble pigments from the flavonoid family known for offering a variety of health benefits [13]. Major anthocyanins, such as cyaniding-3-glucoside, delphinidin-3-glucoside, and petunidin-3-glucoside, have been isolated from the seed coat of black soybeans [18,19]. These compounds are known to alleviate the pathologies of various human diseases, including obesity [16], cardiovascular disorders [20], and cancer [21]. Many metabolomics studies have revealed that the black pigmentation results from the accumulation of anthocyanins in the epidermal palisade layer of the black soybean coat [22,23]. Black soybeans contain abundant isoflavones. Isoflavones are classified into the following 12 chemical forms based on their structures: aglycones (daidzein, glycitein, and genistein), β-glycosides (daidzin, glycitin, and genistin), 6″-*O*-malonylglycosides (malonyldaidzin, malonylglycitin, and malonylgenistin), and 6″-*O*-acetylglycosides (acetyldaidzin, acetylglycitin, and acetylgenistin) [15]. Unlike phenolic compounds and anthocyanins, isoflavones are not uniformly concentrated in black soybeans compared to those in other colored soybeans [24,25,26]. The high anthocyanin and isoflavone contents in black soybeans were demonstrated to have diverse biological activities, including antioxidant properties and health benefits related to cancer, compared to those in other types of soybeans [13]. Furthermore, these compounds may exhibit a range of synergistic effects that can help prevent diseases, such as diabetes, cancer, inflammation, and heart disease [27].

Many researchers have investigated the effects of environmental and genetic factors on the content of anthocyanins, isoflavones, and phenolic compounds across different soybean varieties, including black soybeans [28,29,30,31,32]. These studies have primarily focused on selecting genetic resources to develop soybean cultivars with highly functional components. Similarly, we evaluated the anthocyanin and isoflavone contents of Korean black soybean genotypes in previous studies [14,33] and revealed that Soman possesses the highest isoflavone (i.e., aglycones and glucosides) and anthocyanin (i.e., pelargonidin-3-glucoside) contents. These findings suggest that Soman might possess a greater health functionality effect on the human body compared to that of the other black soybean genotypes. To date, research on the anticancer effects, a key health functionality of black soybeans, has been limited to individual components, such as anthocyanin (i.e., cyanidin 3-*O*-β-d-glucoside) and isoflavone (i.e., daidzein) extracts [34,35,36,37]; however, no research has been conducted on comparing these effects across soybean genotypes. Thus, in this study, we aimed to determine the antitumor activities of Soman using cancer cells and a syngeneic mouse model and elucidate its molecular mechanisms of action. In addition, we compared the antitumor activities between Soman and the Korean black soybean landrace ‘Seoritae’.

In the present study, we show that Soman exhibited superior anticancer activity in vitro and in vivo compared to that of Seoritae. Soman potently inhibited signal transducer and activator of transcription (STAT) phosphorylation, resulting in the decreased expression of glycolytic enzymes (hexokinase 2 [HK2], pyruvate kinase M2 [PKM2], and lactate dehydrogenase A [LDHA]) and activities of pyruvate kinase (PK) and lactate dehydrogenase (LDH), leading to suppressed aerobic glycolysis and proliferation in cancer cells.

## 2. Materials and Methods

### 2.1. Materials

Soman and Seoritae extracts were prepared, as previously described [33]. The STAT1-specific inhibitor, fludarabine (#sc-204755), was purchased from Santa Cruz Biotechnology (Dallas, TX, USA). The ROS scavenger, *N*-Acetyl-l-Cysteine (NAC; #A7250), was purchased from Sigma Aldrich (St. Louis, MO, USA). The antibodies used for immunoblotting and immunohistological analyses are listed in Appendix A.

### 2.2. Cell Culture

B16F10 murine melanoma (#80008), U87MG human glioblastoma (#30014), and MDA-MB-231 human breast carcinoma (#30026) cells were purchased from the Korean Cell Line Bank (Seoul, Republic of Korea). Cancer cells were cultured in Dulbecco’s modified Eagle’s medium (DMEM) (#LM001-05; Welgene, Geongsan, Republic of Korea) supplemented with 10% fetal bovine serum (#S001-04; Welgene) and 1% penicillin–streptomycin (#PS-B; Capricorn Scientific, Ebsdorfergrund, Germany). Normal human astrocytes (NHAs) were kindly provided by Dr. Hyunggee Kim (Korea University, Seoul, Republic of Korea). NHAs were cultured in astrocyte medium (#1801; ScienCell, Carlsbad, CA, USA). All cells were authenticated and routinely tested for mycoplasma.

### 2.3. Immunoblotting Analysis

Immunoblotting analysis was performed, as previously described [38]. Briefly, cells were lysed using a cell lysis buffer (50 mM Tris-HCl [pH 7.5], 150 mM NaCl, 1 mM DTT, 0.5 mM EDTA, 0.1% sodium dodecyl sulfate (SDS), 1% Triton X-100, 100 µM sodium fluoride, 100 µM sodium orthovanadate, 100 µM sodium pyrophosphate, and proteinase inhibitor cocktail). Then, the cell extracts were centrifuged at 15,000 rpm for 15 min, and protein concentrations were measured using the DC protein assay Kit (#5000112; Bio-Rad, Hercules, CA, USA). Equal amounts of protein were resolved using SDS–polyacrylamide gel electrophoresis (PAGE) and thereafter transferred to a nitrocellulose membrane. The membrane was blocked with 5% skim milk at room temperature (RT) for 30 min and then incubated with the indicated antibodies (Appendix A) at 4 °C overnight. The membrane was then incubated with horseradish peroxidase-conjugated secondary antibodies (anti-rabbit [#NA934V; Sigma Aldrich] or anti-mouse [#NA931V; Sigma Aldrich]) at RT for 2 h. Each experiment was repeated at least three times.

### 2.4. Quantitative Real-Time PCR Analysis

Total RNA isolation, reverse transcription, and real-time PCR were performed, as previously described [38]. All reactions were run in triplicate and normalized to the *gapdh* gene. The sequences of primers are summarized in Appendix A.

### 2.5. Cell Proliferation Assay

The cell proliferation assay was performed, as previously described [39]. A total of 4 × 10^4^ cells were seeded onto a 6-well culture plate and counted using a hemocytometer at the indicated times after seeding. For staining, 1 × 10^3^ cells were seeded onto a 96-well culture plate. After four days, the cells were fixed with 10% formalin and stained with 0.1% crystal violet, followed by analysis of the stained area using ImageJ 1.54g software.

### 2.6. Cell Viability Assay to Determine the IC_50_

The half-maximal inhibitory concentration (IC_50_) of the soy extracts was determined based on cell viability. For cell viability measurements, 1 × 10^3^ cells were seeded onto a 96-well culture plate and treated with 0–500 µg/mL soy extracts. The cells were then analyzed using the Quanti-Max WST-8 cell viability assay kit (#QM2500; BIOMAX, Guri, Republic of Korea), according to the manufacturer’s instructions. The IC_50_ values were calculated by plotting cell viability as a function of soy extract concentration using GraphPad Prism software version 9.5.1 (GraphPad Software, Inc., Boston, MA, USA).

### 2.7. Measurement of Glucose Consumption and Lactate Secretion

Glucose consumption and lactate secretion assays were performed, as previously described [40]. Cells were seeded onto a 6-well culture plate and incubated under the indicated conditions. The medium was then harvested for subsequent analyses. To measure glucose consumption, glucose levels were determined using the Glucose Assay Kit (#GAGO20; Sigma Aldrich), according to the manufacturer’s instructions. Glucose consumption was calculated as the difference between the collected and control media. To measure lactate secretion, the collected medium was analyzed using the l-Lactate Assay Kit II (#120005; Eton Bioscience, San Diego, CA, USA), according to the manufacturer’s instructions. All results were normalized to the final cell number.

### 2.8. Measurement of Intracellular ATP Levels and PK and LDH Activity

Cells were seeded onto a 6-well culture plate and incubated under the indicated conditions. ATP levels and PK and LDH activities were determined using the ATP Colorimetric/Fluorometric (#354; BioVision, Milpitas, CA, USA), Pyruvate Kinase Activity Colorimetric/Fluorometric (#K709; BioVision), and Lactate Dehydrogenase Activity Colorimetric (#K726; BioVision) Assay Kits, respectively, according to the manufacturer’s instructions.

### 2.9. Measurement of Intracellular ROS Levels

Intracellular ROS levels were determined as previously described [38]. Cells were seeded onto a 6-well culture plate and incubated under the indicated conditions. The cells were then harvested and analyzed using the DCFDA/H2DCFDA-Cellular ROS Assay Kit (#ab113851; Abcam, Cambridge, MA, USA), according to the manufacturer’s instructions.

### 2.10. Animal Studies

A total of 5 × 10^4^ B16F10 cells were subcutaneously injected into 5-week-old male C57BL/6 mice (OrientBio, Seongnam, Republic of Korea). Three days after tumor implantation, phosphate-buffered saline (PBS) or the indicated extracts were orally administrated daily. The volume of the tumor was calculated using the formula 0.5 × L × W^2^ (L, length; W, width) at the indicated days. Mice were sacrificed 20 days after tumor implantation to analyze the tumor volume and weight, followed by immunohistological analysis. All mice were housed in an animal facility at Dong-A University (Busan, Republic of Korea). All animal experiments were performed in accordance with the relevant institutional and national guidelines. All animal procedures and maintenance conditions were approved by the Institutional Animal Care and Use Committee of Dong-A University (approval number: DIACUC-21-47).

### 2.11. Immunohistological (IHC) Analysis and Scoring

IHC analysis was conducted using paraffin-embedded tumor tissue sections. The expression of phospho-STAT1 and Ki-67 was detected using the VECTASTAIN Elite ABC kit (#PK-6200; Vector Laboratories, Newark, CA, USA) and 3,3′-diaminobenzidine (#SK-4100; Vector Laboratories) and the nuclei were stained with hematoxylin. Six randomly selected fields per slide were analyzed and their results were averaged. Tissue sections were quantitatively evaluated based on the percentage of positive cells and staining intensity, as previously described [40]. Proportion scores were assigned as follows: 0 for 0%, 1 for 0.1–1%, 2 for 1.1–10%, 3 for 11–30%, 4 for 31–70%, and 5 for 71–100% of tumor cells showing positive staining. The staining intensity was rated on a scale of (0–3): 0, negative; 1, weak; 2, moderate; and 3, strong. We then combined the proportion and intensity scores to obtain a total score [range (0−8)], as previously described [41].

### 2.12. Statistical Analysis

All quantitative data are presented as the mean ± SD of at least three independent experiments. Two-group comparisons were conducted using a two-sided, two-sample Student’s *t*-test. SPSS statistical package version 12 (SPSS Inc., Chicago, IL, USA) was used for all statistical analyses. Values of *p* < 0.05 indicated statistically significant differences.

## 3. Results and Discussion

### 3.1. Effects of Soman and Seoritae on the Proliferation of Cancer Cells In Vitro

To determine the antiproliferative effects of Soman and Seoritae on cancer cells, we treated U87MG human glioblastoma, MDA-MB-231 human breast carcinoma, and B16F10 murine melanoma cells with each extract. Black soybeans are known to exert strong antiproliferative effects on many types of cancer cells [13,20,21,42]. As expected, both Soman and Seoritae significantly inhibited the proliferation of all tested cancer cells (Figure 1A,B). Interestingly, Soman exerted more potent antiproliferative effects on cancer cells than Seoritae did. These effects might be due, at least in part, to the higher content of isoflavones and anthocyanins that we had previously evaluated in Soman [33]. However, Soman and Seoritae had no cytotoxic effects on normal human astrocyte (NHA) proliferation (Appendix A), suggesting that the reduction in cell proliferation mediated by Soman and Seoritae is specific to cancer cells. To compare the ability of Soman and Seoritae to inhibit cancer cell proliferation, we treated U87MG, MDA-MB-231, and B16F10 cancer cells with each for 72 h and determined their IC_50_ using the WST-8 assay. As shown in Figure 1C, Soman was more potent than Seoritae in terms of inhibiting the proliferation of cancer cells. Consistent with the cell proliferation data, expression levels of the proliferation markers cyclin D1 and c-Myc were significantly downregulated by Soman, which was greater than the effect observed for Seoritae (Figure 1D). These results indicated that although both Soman and Seoritae could inhibit cancer cell proliferation in vitro, Soman exerted the most potent inhibitory effects.

### 3.2. Effects of Soman and Seoritae on Aerobic Glycolysis in Cancer Cells

To satisfy the demand for rapid proliferation, tumor cells require increased energy. Aerobic glycolysis (or the Warburg effect) is a well-known phenomenon in many types of cancer cells, in which cells exhibit heightened glycolytic activity even in the presence of oxygen, marking a distinct metabolic phenotype [4,5]. Increased glycolytic flux (glucose uptake and lactate production) resulting from aerobic glycolysis supplies cancer cells with energetic and biosynthetic pathways for rapid proliferation [43]. Thus, aerobic glycolysis is critical for cancer cell proliferation, and targeting this process could serve as a new therapeutic strategy for inhibiting tumor cell growth [6,7]. Given that Soman and Seoritae inhibited the proliferation of cancer cells (Figure 1), we considered that the possible mechanism of action involved downregulating aerobic glycolysis to inhibit cancer cell proliferation. Interestingly, the treatment of cancer cells with Soman and Seoritae significantly inhibited glucose consumption (Figure 2A) and lactate secretion (Figure 2B). As expected, total cellular ATP levels were also inhibited in cancer cells after treatment with Soman and Seoritae (Figure 2C). The observed inhibitory effects were more significant in Soman than in Seoritae (Figure 2A–C). These results indicate that Soman and Seoritae could downregulate aerobic glycolysis, which may contribute to the inhibition of cancer cell proliferation, and that Soman exerted the most potent inhibitory effects on aerobic glycolysis.

### 3.3. Effects of Soman and Seoritae on the Phosphorylation of STATs in Cancer Cells

To further investigate the detailed mechanisms by which Soman and Seoritae inhibited aerobic glycolysis in cancer cells, we evaluated the phospho-protein expressional profile of early signals, including ERK, AKT, p38, c-Jun, NF-κB, and STATs, that regulate aerobic glycolysis [44,45,46,47,48,49,50,51]. Notably, treatment with Soman and Seoritae specifically reduced phosphorylation levels of STAT1, 3, and 5, without affecting those of ERK, AKT, p38, c-Jun, and IκBα, in U87MG, MDA-MB-231, and B16F10 cancer cells (Figure 3A). Intriguingly, these observed inhibitory effects were substantially stronger for Soman than for Seoritae. Several studies have demonstrated that specific isoflavones inhibit inflammatory processes by inhibiting STAT1 phosphorylation. For example, genistein inhibits STAT1 phosphorylation in activated macrophages [52]. Similarly, Jantaratnotai et al. reported that genistein and daidzein exerted anti-inflammatory effects against lipopolysaccharide-activated microglia by inhibiting STAT1 phosphorylation [50]. For the first time, we found that the isoflavone-enriched Soman extract inhibited STAT1 phosphorylation in cancer cells. Consistent with previous reports [49,53,54], STAT1 signaling played important roles in aerobic glycolysis and cancer cell proliferation; STAT1 phosphorylation was successfully inhibited by using the STAT1-specific inhibitor, fludarabine (Figure 3B), which resulted in decreased glycolytic enzyme expression, including HK2, PKM2, and LDHA (Figure 3B); PK and LDH activities (Figure 3C); glucose consumption (Figure 3D); lactate secretion (Figure 3E); ATP levels (Figure 3F); and proliferation (Figure 3G) in U87MG and MDA-MB-231 cancer cells.

In addition to STAT1 phosphorylation, we observed for the first time that Soman decreased the phosphorylation of STAT3 and 5 in cancer cells. STAT1, 3, and 5 activation is involved in aerobic glycolysis and proliferation by upregulating the expression of glycolytic enzymes, including HK2 [51,52], PKM2 [55], and LDHA [50,54] (Figure 3B–G). As expected, immunoblotting or quantitative real-time PCR analyses showed that the protein or mRNA expression levels of HK2, PKM2, and LDHA were significantly downregulated after exposure to Soman and Seoritae, where Soman exerted the most potent inhibitory effect on all tested cancer cells (Figure 3H). Consistent with the inhibitory effects observed for PKM2 and LDHA expression, the enzyme activities of PK and LDH were decreased after treatment with Soman and Seoritae (Figure 3I). These results indicate that both Soman and Seoritae could decrease the phosphorylation of STAT1, 3, and 5, which downregulates aerobic glycolysis and proliferation in cancer cells; these effects were more significant for Soman than for Seoritae.

Soybean-derived bioactive components, including isoflavones, have been shown to induce anticancer effects through a reactive oxygen species (ROS)-dependent pathway in various cancer types [56,57,58,59,60,61]. Treatment with Soman and Seoritae significantly increased intracellular ROS production in all tested cancer cells, with Soman inducing a greater increase in ROS levels compared to Seoritae (Figure 3J). Additionally, the expression levels of nuclear factor-erythroid 2 related factor 2 (Nrf2), a key regulator of antioxidant response elements [62], and its downstream target gene solute carrier family 7 member 11 (SLC7A11) [63] were upregulated following treatment with Soman and Seoritae (Figure 3K). We next investigated the involvement of ROS in the anticancer effects of Soman and Seoritae. As shown in Figure 3L, blocking ROS generation (Figure 3J) using the ROS scavenger *N*-acetyl-l-cysteine (NAC) partially restored Soman- or Seoritae-reduced cancer cell viabilities. However, the reduced phosphorylation levels of STAT1, 3, and 5 by Soman and Seoritae were not affected by NAC treatment (Figure 3K). These findings suggest that Soman and Seoritae inhibit STAT phosphorylation in an ROS-independent manner and exert anticancer activities through both ROS-dependent and -independent pathways.

### 3.4. Effects of Soman and Seoritae on Tumor Growth In Vivo

Finally, to investigate whether Soman and Seoritae exerted potent antitumor growth effects in vivo, B16F10 murine melanoma cells were subcutaneously implanted into syngeneic C57BL/6 mice, followed by the oral administration of Soman or Seoritae (Figure 4A). No abnormalities in mouse body weight or behavior were observed. Both Soman and Seoritae effectively exhibited antitumor activity in B16F10 cell-harboring mice, as reflected by a decrease in tumor volume and weight (Figure 4B–D). As expected, we observed that Soman exerted the most potent inhibitory effect on B16F10 tumor growth (Figure 4B–D). IHC analysis further revealed that cancer cell proliferation, as evidenced by the Ki-67 expression and STAT1 phosphorylation levels measured, was considerably decreased in Soman-treated mice compared to that in Seoritae-treated mice (Figure 4E), which is consistent with the in vitro results. These results indicate that both Soman and Seoritae inhibited tumor growth in vivo, with Soman exerting the most potent suppressive effect.

In summary, our results showed that Soman exhibited enhanced antitumor activities compared to those of the black soybean landrace; it decreased the phosphorylation of STATs, which may downregulate aerobic glycolysis and cell proliferation, leading to the inhibition of tumor growth. In the present study, we focused on the effect of Soman on STAT-mediated aerobic glycolysis. However, we could not exclude the possibility that the aerobic glycolysis-independent pathway could also be involved in the observed antitumor activities because STAT signals participate in numerous cellular processes to promote tumor growth [64,65]. In addition, comprehensive mechanisms underlying the Soman-induced inhibition of STAT phosphorylation (an ROS-independent pathway) and the types of components in Soman responsible for this will be worth exploring in future studies.

## 4. Conclusions

Our study findings demonstrate that Soman exhibits superior anticancer activities in vitro and in vivo compared to those of the black soybean landrace Seoritae. Soman effectively inhibited STAT phosphorylation in an ROS-independent manner, resulting in the decreased expression of glycolytic enzymes (HK2, PKM2, and LDHA) and activities of PK and LDH, leading to the suppression of aerobic glycolysis, cancer cell proliferation, and tumor growth. Our findings suggest that the dysregulation of glucose metabolism could serve as a new mechanistic foundation for understanding the anticancer activities of Soman, which could offer therapeutic potential for treating patients with cancer.

## Figures and Tables

**Figure 1 antioxidants-14-00228-f001:**
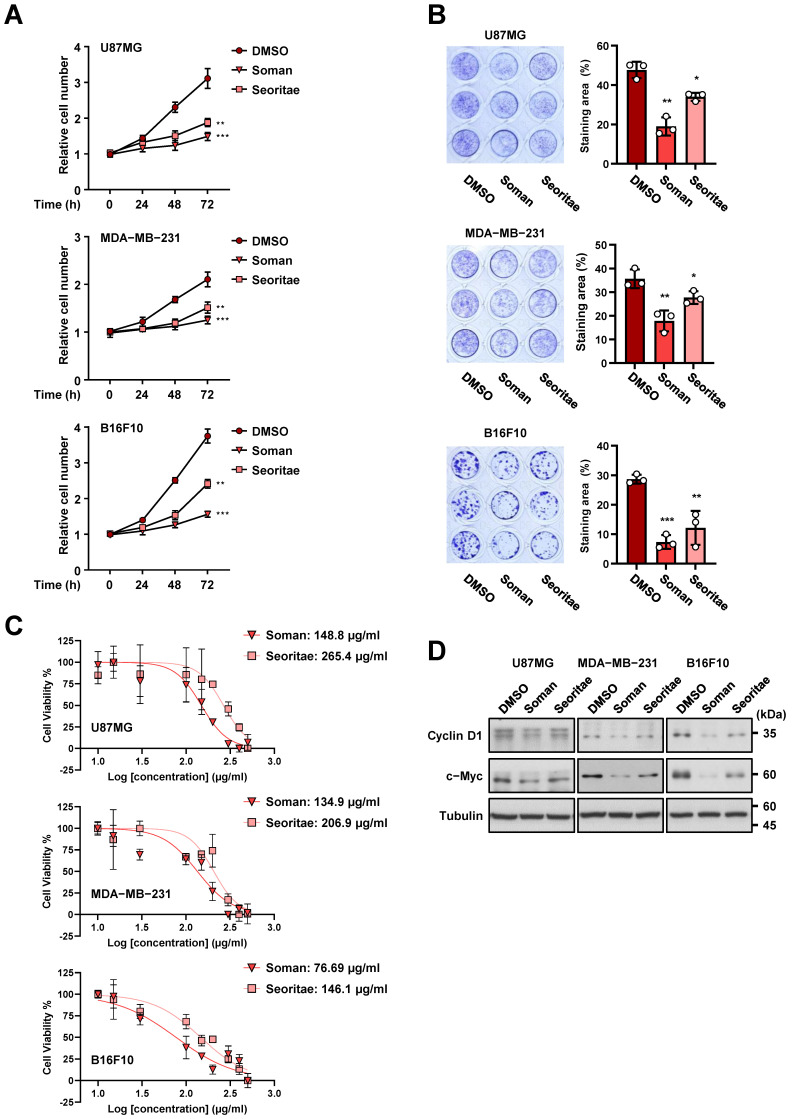
Effects of Soman and Seoritae on the proliferation of cancer cells in vitro. (**A**) U87MG, MDA-MB-231, and B16F10 cells were cultured with or without the indicated soybean extracts for the indicated periods of time and thereafter harvested for cell counting. (**B**) U87MG, MDA-MB-231, and B16F10 cells were cultured with or without the indicated soybean extracts. The colonies were fixed with 10% formalin and stained with 0.1% crystal violet. Representative images (**left** panel) and graphs of the staining area (**right** panel) are shown. (**C**) U87MG, MDA-MB-231, and B16F10 cells were cultured with or without the indicated soybean extracts for three days. Cell viability was measured using the WST-8 viability assay and the IC_50_ was calculated using nonlinear regression analysis. (**D**) U87MG, MDA-MB-231, and B16F10 cells were cultured with or without the indicated soybean extract for one day and immunoblotting analyses were performed with the indicated antibodies. Data are presented as the mean ± SD of three independent experiments (**A**–**C**). * *p* < 0.05; ** *p* < 0.01; *** *p* < 0.001, based on Student’s *t*-test.

**Figure 2 antioxidants-14-00228-f002:**
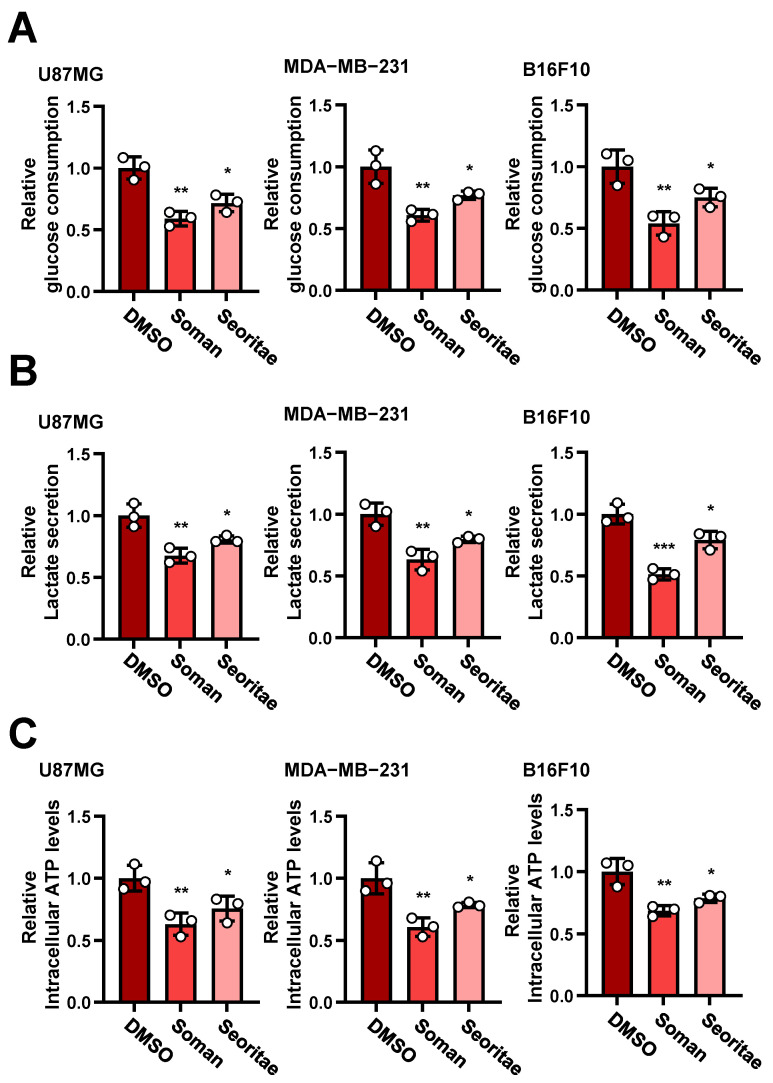
Effects of Soman and Seoritae on aerobic glycolysis in cancer cells. (**A**,**B**) U87MG, MDA-MB-231, and B16F10 cells were cultured with or without the indicated soybean extract for two days. The media were collected to analyze glucose consumption (**A**) and lactate secretion (**B**). (**C**) U87MG, MDA-MB-231, and B16F10 cells were cultured with or without the indicated soybean extract for one day. The cells were collected to analyze intracellular ATP levels. Data are presented as the mean ± SD of three independent experiments (**A**–**C**). * *p* < 0.05; ** *p* < 0.01; *** *p* < 0.001, based on Student’s *t*-test.

**Figure 3 antioxidants-14-00228-f003:**
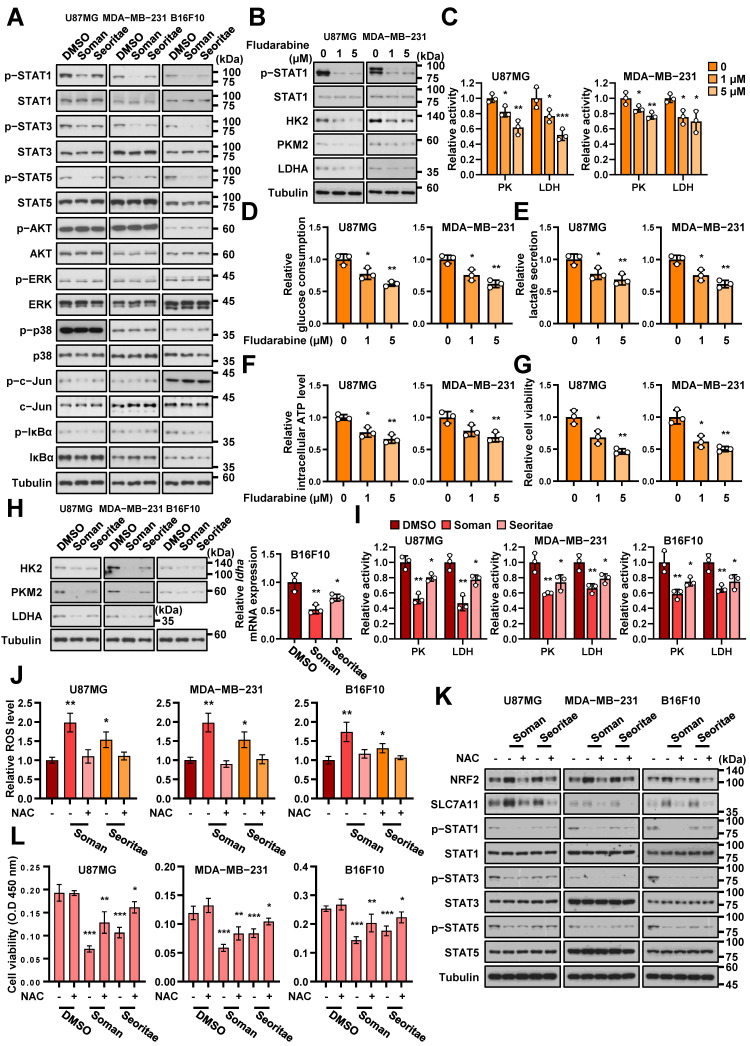
Effects of Soman and Seoritae on the phosphorylation of STATs in cancer cells. (**A**) U87MG, MDA-MB-231, and B16F10 cells were cultured with or without the indicated soybean extracts for one day and immunoblotting analyses were performed with the indicated antibodies. (**B**,**C**) U87MG and MDA-MB-231 cells were treated with or without fludarabine at the indicated concentrations for one day and immunoblotting analyses were performed with the indicated antibodies (**B**). The pyruvate kinase (PK) and lactate dehydrogenase (LDH) activities were assessed (**C**). (**D**,**E**) U87MG and MDA-MB-231 cells were treated with or without fludarabine at the indicated concentrations for two days. The medium was then collected to analyze glucose consumption (**D**) and lactate secretion (**E**). (**F**) U87MG and MDA-MB-231 cells were treated with or without fludarabine at the indicated concentrations for two days and the cells were collected to analyze intracellular ATP levels. (**G**) U87MG and MDA-MB-231 cells were treated with or without fludarabine at the indicated concentrations and time points. Cell viability was measured using the WST-8 viability assay. (**H**) U87MG, MDA-MB-231, and B16F10 cells were cultured with or without the indicated soybean extracts for one day and immunoblotting or quantitative real-time PCR analyses were performed with the indicated antibodies. (**I**) U87MG, MDA-MB-231, and B16F10 cells were cultured with or without the indicated soybean extracts for one day, followed by analyses of the PK and LDH activities. (**J**–**L**) U87MG, MDA-MB-231, and B16F10 cells were cultured with or without the indicated soybean extracts or NAC (2 mM) for one day, followed by analyses of intracellular ROS levels (**J**), immunoblotting (**K**), and cell viability (**L**). Data are presented as the mean ± SD of three independent experiments (**C**–**J**,**L**). * *p* < 0.05; ** *p* < 0.01; *** *p* < 0.001, based on Student’s *t*-test.

**Figure 4 antioxidants-14-00228-f004:**
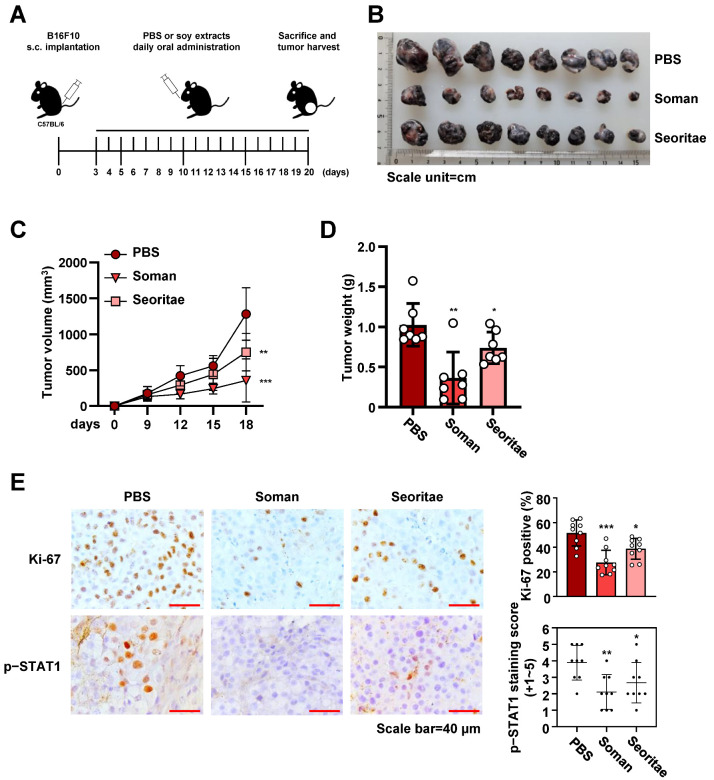
Effects of Soman and Seoritae on tumor growth in vivo. (**A**) Schematic diagram of the in vivo experimental procedures. B16F10 cells were subcutaneously injected into C57BL/6 mice. After three days of cell injection, the mice received oral administration of soy extracts or phosphate-buffered saline (PBS) daily. (**B**) Representative images of B16F10 tumors. Scale unit = cm. (**C**) Growth curve of tumor burden derived from the indicated soy extract-treated or PBS-treated mice. (**D**) Tumor weight of tumors on day 20 derived from sacrificed mice. (**E**) Representative immunohistochemical micrographs of the analyzed tumor tissues derived from sacrificed mice with the indicated antibodies (**left** panel), with the quantification graph (**right** panel) shown. Data are presented as the mean ± SD of three independent experiments (**C**–**E**). * *p* < 0.05; ** *p* < 0.01; *** *p* < 0.001, based on Student’s *t*-test.

## Data Availability

Data are contained within the article or Appendix A.

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
