# Peer review of "Enhanced Antitumor Activity of Korean Black Soybean Cultivar ‘Soman’ by Targeting STAT-Mediated Aerobic Glycolysis"

_antioxidants, 2025, doi:10.3390/antiox14020228_

Round 1

Reviewer 1 Report (Previous Reviewer 2)

Dear Author,

I read the revised version of the manuscript and agree that oxidative stress is involved in the cytostatic activity of your extracts. As you showed, the effect on STAT1 is ROS-independent, which seems to confirm that other mechanisms are also involved. I wondered whether the effects come from isoflavones or other compounds but you justified this by literature data and I can accept it. I still do not feel that the Antioxidants are the best place for that manuscript but the role of ROS-induced signaling now is partially confirmed. 

I do not have any. 

Author Response

Major comments

I read the revised version of the manuscript and agree that oxidative stress is involved in the cytostatic activity of your extracts. As you showed, the effect on STAT1 is ROS-independent, which seems to confirm that other mechanisms are also involved. I wondered whether the effects come from isoflavones or other compounds but you justified this by literature data and I can accept it. I still do not feel that the Antioxidants are the best place for that manuscript but the role of ROS-induced signaling now is partially confirmed.

Response: We sincerely appreciate the reviewer’s thoughtful feedback and their acknowledgment of the role of oxidative stress in the cytostatic activity of our extracts, as well as the ROS-independent effects on STATs signaling. We are also grateful for their acceptance of our justification regarding the involvement of isoflavones or other compounds, supported by literature data. Regarding the concern about the suitability of Antioxidants journal for this manuscript, we respectfully emphasize that the central theme of our study aligns well with the journal’s scope. Our work not only highlights the role of ROS-induced signaling in the extracts-induced cytostatic effects but also explores the involvement of ROS on molecular pathway affected by the extracts, which we believe will be of interest to the readership of Antioxidants journal. Thank you once again for your valuable insights, which have significantly improved the quality of our work.

Detail comments

I do not have any.

Response: Thanks.

Reviewer 2 Report (Previous Reviewer 1)

Manuscript ID: antioxidants-3474066

The manuscript by Su Hwan Park et al. is very interesting and well-written and describes the anticancer effects of the natural extracts obtained from the Soman and Seoritae soybean cultivars in three different cancer cell lines (B16F10, U87MG, and MDA-MB-231) and also in the B16F10 murine melanoma syngeneic in vivo model. Furthermore, the authors demonstrated that these natural extracts (in particular the Soman extract) inhibited the aerobic glycolysis molecular pathways of the tumor cells and significantly reduced the levels of p-STAT1, p-STAT3, and p-STAT5 transcription factors, which play a pivotal role in the regulation of the markers of the aerobic glycolysis molecular pathway. Furthermore, the authors showed that the treatments with Seoritae and Soman extracts increased the Nrf2 and SLC7A11 levels. The relevance of this scientific contribution is represented by the fact that the authors investigated the modulation of the biomarkers of an important hallmark of cancer which is less studied when compared with the hallmarks focused on the resistance to pro-apoptotic mechanisms and on the evasion of growth suppressors by cancer cells. In particular, the authors identified p-STAT1 as an in vivo biomarker that should be inhibited to reduce the proliferation rate and the growth of tumors. 

The authors should answer my minor requests to further improve their Manuscript.

MINOR REVISIONS

1)Figure 3: The authors showed that the Seoritae and Soman natural extracts significantly increased the levels of ROS in U87MG, MDA-MB-231, and B16F10 cells, indicating a pro-oxidant effect. In addition, the authors showed an increase in the levels of both Nrf2 (the transcription factor that activates the antioxidant mechanisms of the cells) and its target SLC7A11, indicating an antioxidant effect. How do the authors explain these contradictory results? Please describe the molecular pathway modulated by the Seoritae and Soman natural extracts which could explain both these experimental results.

Author Response

Major comments

The manuscript by Su Hwan Park et al. is very interesting and well-written and describes the anticancer effects of the natural extracts obtained from the Soman and Seoritae soybean cultivars in three different cancer cell lines (B16F10, U87MG, and MDA-MB-231) and also in the B16F10 murine melanoma syngeneic in vivo model. Furthermore, the authors demonstrated that these natural extracts (in particular the Soman extract) inhibited the aerobic glycolysis molecular pathways of the tumor cells and significantly reduced the levels of p-STAT1, p-STAT3, and p-STAT5 transcription factors, which play a pivotal role in the regulation of the markers of the aerobic glycolysis molecular pathway. Furthermore, the authors showed that the treatments with Seoritae and Soman extracts increased the Nrf2 and SLC7A11 levels. The relevance of this scientific contribution is represented by the fact that the authors investigated the modulation of the biomarkers of an important hallmark of cancer which is less studied when compared with the hallmarks focused on the resistance to pro-apoptotic mechanisms and on the evasion of growth suppressors by cancer cells. In particular, the authors identified p-STAT1 as an in vivo biomarker that should be inhibited to reduce the proliferation rate and the growth of tumors.

Response: Thanks.

Detail comments

The authors should answer my minor requests to further improve their Manuscript.

MINOR REVISIONS

1)Figure 3: The authors showed that the Seoritae and Soman natural extracts significantly increased the levels of ROS in U87MG, MDA-MB-231, and B16F10 cells, indicating a pro-oxidant effect. In addition, the authors showed an increase in the levels of both Nrf2 (the transcription factor that activates the antioxidant mechanisms of the cells) and its target SLC7A11, indicating an antioxidant effect. How do the authors explain these contradictory results? Please describe the molecular pathway modulated by the Seoritae and Soman natural extracts which could explain both these experimental results.

Response: The Reviewer’s point is well-taken. The transcription factor Nrf2 is the master regulator of the cellular antioxidant response. Nrf2 is responsible for regulating an extensive panel of antioxidant enzymes involved in the detoxification and elimination of oxidative stress. It is well known that elevated ROS levels enhance Nrf2 transcriptional activity and protein stability to mitigate oxidative damage (PMID: 15572695, 23404377, 23849864, 34803685, 36552553). Hence, the upregulation of both Nrf2 and its target, SLC7A11, in response to increased ROS is a common cellular antioxidant defense mechanism induced by black soybean extracts.

This manuscript is a resubmission of an earlier submission. The following is a list of the peer review reports and author responses from that submission.

Round 1

Reviewer 1 Report

Manuscript ID: antioxidants-3305707

The manuscript by Jeong Hyun Seo et al. is very interesting and well-written and describes the anticancer effects of the natural extracts obtained from the Soman and Seoritae soybean cultivars in three different cancer cell lines (B16F10, U87MG, and MDA-MB-231) and also in the B16F10 murine melanoma syngeneic in vivo model. Furthermore, the authors demonstrated that these natural extracts (in particular the Soman extract) inhibited the aerobic glycolysis molecular pathways of the tumor cells and significantly reduced the levels of p-STAT1, p-STAT3, and p-STAT5 transcription factors, which play a pivotal role in the regulation of the markers of the aerobic glycolysis molecular pathway. The relevance of this scientific contribution is represented by the fact that the authors investigated the modulation of the biomarkers of an important hallmark of cancer which is less studied when compared with the hallmarks focused on the resistance to pro-apoptotic mechanisms and on the evasion of growth suppressors by cancer cells. In particular, the authors identified p-STAT1 as an in vivo biomarker that should be inhibited to reduce the proliferation rate and the growth of tumors. The authors should answer my major and minor requests to improve their Manuscript. 

Manuscript ID: antioxidants-3305707

MAJOR REVISION

1)The authors showed that the Soman and Seoritae natural extracts reduced the cell viability (%) of the U87MG, MDA-MB-231, and B16F10 cancer cells, but they should also demonstrate that these treatments do not exert a significant cytotoxic effect in normal non-tumorigenic cells like, for example, Primary Epidermal Melanocytes (HEMa cells). These new results will demonstrate that the Soman and Seoritae-mediated reduction of cell viability (%) is specific for cancer cell lines.

MINOR REVISIONS

1)Figure 1D: The authors should provide a better western blot image showing the reduction of c-Myc levels in MDA-MB-231 cancer cells mediated by Soman and Seoritae natural extracts.

2)Figure 3A: The authors should provide a better western blot image showing the levels of p-p38 in U87MG cancer cells after the treatment with Soman and Seoritae extracts.

3)Figure 3H: The authors should provide the western blot image showing the levels of LDHA in B16F10 cancer cells after the treatment with Soman and Seoritae natural extracts.

TYPOS

In the Conclusions section, the authors should remove the word “strongly” from the sentence “Our study findings strongly demonstrate that…”.

Author Response

#Reviewer 1

Manuscript ID: antioxidants-3305707

MAJOR REVISION

1)The authors showed that the Soman and Seoritae natural extracts reduced the cell viability (%) of the U87MG, MDA-MB-231, and B16F10 cancer cells, but they should also demonstrate that these treatments do not exert a significant cytotoxic effect in normal non-tumorigenic cells like, for example, Primary Epidermal Melanocytes (HEMa cells). These new results will demonstrate that the Soman and Seoritae-mediated reduction of cell viability (%) is specific for cancer cell lines.

Response: The Reviewer’s point is well-taken. To evaluate whether Soman and Seoritae natural extracts exert cytotoxic effects on normal non-tumorigenic cells, we conducted experiments using normal human astrocytes (NHA). As shown in revised Supplemental Figure S1, Soman and Seoritae had no cytotoxic effects on NHA proliferation. These findings suggest that the reduction in cell proliferation mediated by Soman and Seoritae is specific to cancer cell lines. We have included the result in the main text of the revised manuscript.

MINOR REVISIONS

1)Figure 1D: The authors should provide a better western blot image showing the reduction of c-Myc levels in MDA-MB-231 cancer cells mediated by Soman and Seoritae natural extracts.

Response: We thank the reviewer for pointing this out. We have provided a better western blot image of c-Myc in the revised Figure 1D.

2)Figure 3A: The authors should provide a better western blot image showing the levels of p-p38 in U87MG cancer cells after the treatment with Soman and Seoritae extracts.

Response: We thank the reviewer for pointing this out. We have provided a better western blot image of p-p38 in the revised Figure 3A.

3)Figure 3H: The authors should provide the western blot image showing the levels of LDHA in B16F10 cancer cells after the treatment with Soman and Seoritae natural extracts.

Response: We noticed that while the antibodies used for HK2 and PKM2 are reactive to both human and mouse species, the anti-LDHA antibody employed is specific to human LDHA protein only. Thus, we have checked mRNA expression levels of LDHA in B16F10 murine melanoma cells. Consistent with the observed LDHA mRNA expression, LDH enzyme activity was reduced following treatment with Soman and Seoritae, likely due to a decrease in LDHA protein expression.

TYPOS

In the Conclusions section, the authors should remove the word “strongly” from the sentence “Our study findings strongly demonstrate that…”.

Response: As suggested, we have removed the word “strongly” in the conclusions section.

Reviewer 2 Report

The study is very interesting but antioxidant properties are mentioned only in the Introduction in the context of the antioxidant potential of soybeans. There are no studies of antioxidant properties in the manuscript which is focused on anticancer properties in vitro and in vivo (also a valuable analysis mode of action). Of course, the study is based on the previous one but it is unclear what kind of extract was studied and if antioxidants are present [ref. 33- 3 different extraction methods). Moreover, those analyses do not include changes in antioxidant potential and thus do not cover any aims and scopes of the journal.  In my opinion, the study does not fit the scope of The Antioxidant. I strongly recommend some other journals e.g. Molecules, Nutrients. 

See above

Author Response

#Reviewer 2

The study is very interesting but antioxidant properties are mentioned only in the Introduction in the context of the antioxidant potential of soybeans. There are no studies of antioxidant properties in the manuscript which is focused on anticancer properties in vitro and in vivo (also a valuable analysis mode of action). Of course, the study is based on the previous one but it is unclear what kind of extract was studied and if antioxidants are present [ref. 33- 3 different extraction methods). Moreover, those analyses do not include changes in antioxidant potential and thus do not cover any aims and scopes of the journal. In my opinion, the study does not fit the scope of The Antioxidant. I strongly recommend some other journals e.g. Molecules, Nutrients.

Response: We sincerely appreciate your thoughtful comments and the opportunity to clarify how our study aligns with the scope of the Antioxidants journal, particularly the Special Issue "Plant Antioxidants, Inflammation, and Chronic Disease."

We selected this Special Issue for submission because our study investigates the anticancer effects of black soybean extracts (Soman and Seoritae), which are strongly linked to their antioxidant properties. Black soybeans are widely recognized for their roles in mitigating oxidative stress, inflammation, and chronic diseases, including cancer. As outlined in our manuscript, black soybeans are rich in anthocyanins and isoflavones—potent antioxidants. These compounds are known to modulate oxidative stress, a critical driver of chronic inflammation and cancer progression. This aligns closely with the focus of the Special Issue.

Our research builds upon previous findings (Ref. 33) demonstrating the superior antioxidant potential of Soman compared to other black soybean cultivars. Although this manuscript primarily explores the anticancer mechanisms of Soman and Seoritae, the observed effects are inherently connected to their antioxidant activities. Compounds from Soman black soybean not only quench reactive oxygen species but also influence cellular signaling pathways, such as STAT phosphorylation, which play pivotal roles in cancer development and progression. These mechanistic insights bridge the fields of antioxidants and chronic disease research.

The extracts used in this study were prepared following methods detailed in Ref. 33, which extensively analyzed the antioxidant activities of these extracts. While we did not directly measure antioxidant activity in this manuscript, the presence of anthocyanins and isoflavones supports the hypothesis that their antioxidant potential contributes to the anticancer effects we observed.

Therefore, we respectfully suggest that the manuscript aligns well with the aims of the Antioxidants journal and the Special Issue by contributing novel insights into the relationship between plant antioxidants and chronic diseases like cancer.

Round 2

Reviewer 1 Report

The manuscript by Jeong Hyun Seo et al. is very interesting and well-written and describes the anticancer effects of the natural extracts obtained from the Soman and Seoritae soybean cultivars in three different cancer cell lines (B16F10, U87MG, and MDA-MB-231) and also in the B16F10 murine melanoma syngeneic in vivo model. Furthermore, the authors demonstrated that these natural extracts (in particular the Soman extract) inhibited the aerobic glycolysis molecular pathways of the tumor cells and significantly reduced the levels of p-STAT1, p-STAT3, and p-STAT5 transcription factors, which play a pivotal role in the regulation of the markers of the aerobic glycolysis molecular pathway. The relevance of this scientific contribution is represented by the fact that the authors investigated the modulation of the biomarkers of an important hallmark of cancer which is less studied when compared with the hallmarks focused on the resistance to pro-apoptotic mechanisms and on the evasion of growth suppressors by cancer cells. In particular, the authors identified p-STAT1 as an in vivo biomarker that should be inhibited to reduce the proliferation rate and the growth of tumors. The authors should answer my minor requests to further improve their Manuscript. 

MINOR REVISIONS

The authors answered all my minor requests and improved their manuscript. Furthermore, the authors tried to answer my major revision, investigating the effects of the studied natural extracts in normal human astrocytes (NHA). Unluckily, when I opened the file corresponding to the "Supplemental material", I found only the "Supplemental Table 1", "Supplemental Table 2" and the Caption of the "Supplemental Figure 1", but I did not find the images of the "Supplemental Figure 1", which should show that the investigated natural extracts do not exert a significant cytotoxic effect in NHA cells. Please provide the PDF file of the New "Supplemental Figure 1" with also the images in an improved version of the "Supplemental material file".

Reviewer 2 Report

Dear authors,

I have read your explanation regarding my comments and reviewed the manuscript again. I understand that soybeans are a source of isoflavones, but I am not convinced that these compounds are responsible for the observed anticancer effects. Perhaps introducing additional controls in the study, containing soybean isoflavones, would allow you to justify that these compounds are responsible for the indicated activities. As I said the studies are valuable but do not fit the scope. In light of the above, I maintain my decision.

see above